# Biomolecular and Genetic Prognostic Factors That Can Facilitate Fertility-Sparing Treatment (FST) Decision Making in Early Stage Endometrial Cancer (ES-EC): A Systematic Review

**DOI:** 10.3390/ijms23052653

**Published:** 2022-02-28

**Authors:** Panayiotis Tanos, Savvas Dimitriou, Giuseppe Gullo, Vasilios Tanos

**Affiliations:** 1Institute of Applied Health Sciences, University of Aberdeen & Aberdeen Royal Infirmary, Aberdeen AB25 2ZN, UK; 2Aberdeen Fertility Centre, NHS Grampian and University of Aberdeen, Aberdeen AB25 2ZN, UK; savvas.dimitrious@nhs.scot; 3In Vitro Fertilization Unit (IVF Unit), Azienda Ospedaliera Ospedali Riuniti, Villa Sofia Cervello, 90146 Palermo, Italy; gullogiuseppe@libero.it; 4Department of Obstetrics and Gynecology, Aretaeio Hospital, Nicosia 2024, Cyprus; v.tanos@aretaeio.com; 5St. Georges’ Medical School, University of Nicosia, Nicosia 2408, Cyprus

**Keywords:** early stage endometrial cancer, conservative treatment, fertility-sparing surgery, fertility-sparing treatment, reproductive age, genetic prognostic factors, biomolecular prognostic factors

## Abstract

Endometrial cancer occurs in up to 29% of women before 40 years of age. Seventy percent of these patients are nulliparous at the time. Decision making regarding fertility preservation in early stage endometrial cancer (ES-EC) is, therefore, a big challenge since the decision between the risk of cancer progression and a chance to parenthood needs to be made. Sixty-two percent of women with complete remission of ES-EC after fertility-sparing treatment (FST) report to have a pregnancy wish which, if not for FST, they would not be able to fulfil. The aim of this review was to identify and summarise the currently established biomolecular and genetic prognostic factors that can facilitate decision making for FST in ES-EC. A comprehensive search strategy was carried out across four databases; Cochrane, Embase, MEDLINE, and PubMed; they were searched between March 1946 and 22nd December 2022. Thirty-four studies were included in this study which was conducted in line with the PRISMA criteria checklist. The final 34 articles encompassed 9165 patients. The studies were assessed using the Critical Appraisal Skills Program (CASP). *PTEN* and *POLE* alterations we found to be good prognostic factors of ES-EC, favouring FST. MSI, *CTNNB1*, and *K-RAS* alterations were found to be fair prognostic factors of ES-EC, favouring FST but carrying a risk of recurrence. *PIK3CA*, *HER2*, *ARID1A*, *P53*, *L1CAM*, and *FGFR2* were found to be poor prognostic factors of ES-EC and therefore do not favour FST. Clinical trials with bigger cohorts are needed to further validate the fair genetic prognostic factors. Using the aforementioned good and poor genetic prognostic factors, we can make more confident decisions on FST in ES-EC.

## 1. Introduction

According to the World Health Organization, endometrial carcinoma (EC) is the most common gynaecological cancer in Europe [1]. The incidence of EC is approximately 15,000 newly diagnosed women each year, of which 4% are at reproductive age [2]. In low-risk diseases, total hysterectomy and bilateral salpingo-oophorectomy provide patients with up to 93% chance of cure [3]. However, temporal preservation of the uterus in early stage endometrial cancer (ES-EC) is an available option for women who have a strong will to preserve fertility and achieve spontaneous pregnancy. This raises primarily medical but also ethical and social dilemmas since fertility-sparing treatment (FST) consists of compromising radical care in the effort of allowing them to reproduce. Sometimes, these patients will need to preserve their fertility by assisted reproductive technology (ART) [4]. They would also freeze their oocytes via a vitrification system, usually using a GnRh antagonist as a trigger to freeze all the gametes in ovarian stimulation [5,6,7].

Type 1, low-grade EC (G1, G2), is considered an ES-EC. It is currently the only histological type of EC that can be addressed with a fertility-sparing approach. For FST to be possible, myometrium and lymph-vascular space must not be involved adnexal invasion should not be seen. Preliminary evidence in disease progression and life expectancy in patients following temporal uterine preservation for ES-EC are encouraging and appear to be an acceptable management option. In a recent systematic review by Schuurman et al., 2021 [8] 62.6% of patients with complete remission on FST were reported to have a pregnancy wish. Among these patients with complete remission, 36.9% became pregnant. Nevertheless, the long-term outcome, survival rate, and quality of life in these patients are not yet prospectively investigated.

Most EC cases are sporadic, and only 10% of them are considered familiar. For that small percentage of usually younger individuals, tissue genetics and biomolecular markers are vital prognostic factors for the extent of EC progression, and it is, therefore, important to consider them prior to the FST decision. A diagnostic classification of the tumour based on molecular biology was provided by The Cancer Genome Atlas Research Network (CGARN) in 2013. CGARN prompted a growing interest in risk factor stratification of patients based on molecular biology and genetics of the tumour [9]. Some of these prognostic factors, microsatellite instability (MSI), mismatch repair genes (MMR), polymerase epsilon (*POLE*), tumour protein 53 (*TP53*), human epidermal growth factor receptor 2 (*HER2*), Kirsten rat sarcoma viral oncogene homolog (KRAS) and phosphatase, and tensin homolog (*PTEN*) mutations are already well established in the clinical setting and management of patients. Further research in recent years continued to support these as prognostic factors and gave way for the novel discovery of additional genetic and biomolecular markers with promising results.

There are several reasons why the identification and validation of prognostic factors is important in oncology [10]. By determining which genes or biomolecular factors are prognostic of outcomes, we gain insights into the physiology as well as pathology of the disease. Secondary, appropriate treatment modalities can be established either through genetically targeted treatments or through treatment personalised to the patient. Prognostic factors can also be used in the design, conduct, and analysis of clinical trials as well as preventatively for patients’ families and informatively regarding their own risk of recurrence or death [11].

The aim of this review is to identify and summarise the currently established biomolecular and genetic prognostic factors that can facilitate the decision for FST in cases of ES-EC. Markers that designate bad prognosis, metastasis, and early recurrency could be used to deny FST. On the other hand, markers that demonstrate a good prognosis can help clinicians in decision-making for the management of patients wishing to preserve fertility. The secondary outcome was setting an initial path towards establishing guidelines for FST management in patients with ES-EC. In doing so, without forgetting that an individual approach is mandatory as each patient’s characteristics and expectations regarding motherhood differ, it is equally important for the psychological impact of these gynaecological diseases to be considered [12].

## 2. Results

### 2.1. Endometrioid Endometrial Cancer (EECs)

Bokhman et al., 1987 identified 70–80% of EC as EECs. EECs are linked to unopposed estrogen stimulation in young postmenopausal women [13]. Jongen et al., 2009 [14] conclude that while patients with estrogen receptor-alpha positive tumours have better overall survival, the absence of progesterone receptor-A is also an independent prognostic factor for disease-free survival and disease relapse. Specifically, the expression of estrogen receptor-A and the ratio of progesterone receptor-A and -B were associated with lower grade tumours as well as both shorter disease-free survival and shorter overall survival.

EECs can be further differentiated according to clinical and histopathological variables but also according to the activation and inactivation of certain genes. Studies showed that EECs with *K-RAS*, *HER2*, and b-Catenin gain function as well as microsatellite and *PTEN* loss-of-function alterations [15,16,17]. Other commonly mutated genes in ES-EC include *FGFR2*, *ARID1A*, *CTNNB1*, *PIK3CA*, and PIK3R1 [18,19,20]. However, the benefit of the classification of the aforementioned genes in premenopausal women who are interested in FST is unclear [21].

### 2.2. Established Genetic and Biomolecular Markers as Prognostic Factors

Four prognostic categories were established through CGARN. *POLE* ultra-mutated, MSI hypermutated low copy-number abnormalities, and high copy-number abnormalities. Each group is characterised by specific mutations and a different prognosis (Figure 1).

The *POLE* ultra-mutated category has the most favourable prognosis. It is currently associated with longer progression-free survival and correlated with *PTEN*, PIK3R1, *PIK3CA*, FBXW7, KRAS, and *TP53* mutated genes [17]. MSI and MMR groups are portrayed to have an intermediate prognosis. The specific mutated genes involved with this group are *PTEN*, KRAS, and *ARID1A* [22]. The MSI group represents altered mechanisms of MMR genes (MLH1, MSH2, MSH6, or PMS2), of which their inactivation leads to MSI accumulations [23]. Finally, we have the low copy number group. This group has an intermediate prognosis and is associated with *CTNNB1* and *PTEN* gene alterations [22]. The high copy-number group is linked to high-grade EC and, more specifically, the serous histotype; therefore, we are not going to discuss this further. A more detailed analysis is presented in Table 1 and will be described further below. Figure 2 portray a visual representation of the functions of each gene described and the pathological consequences of their mutations.

#### 2.2.1. *PTEN*

The phosphatase and tensin homolog (*PTEN*) mutations occur early in the neoplastic process of ES-EC reported in 57−83% of cases and represent the most common genetic mutation reported [48]. *PTEN* gene alteration is located at chromosome 10q23 and behaves as a tumour suppressor gene. It encodes for both a lipid and a protein phosphatase, inducing cell cycle arrest at the G1/S checkpoint. Additionally, it inhibits the growth-factor-stimulated mitogen-activated protein kinase (MAPK) signalling pathway. Subsequently, it affects focal adhesion formation, cellular differentiation, and proliferation as well as cell spread, migration, inflammatory responses, and apoptosis [49]. Salvesen et al., 2004 [24] demonstrated a significant (*p* = 0.05) association between *PTEN* expression loss and metastatic disease. *PTEN* mutation in exon 5 and 8 was also significantly correlated with ES-EC, low grade, young age, and favourable prognosis. Additionally, *PTEN* alterations were associated with microsatellite instability (MSI), decreased hMLH1 expression, hMLH1 inactivation, and hMLH1 methylation [24].

*PTEN* negatively regulates the downstream pathway of phosphatidylinositol 3-kinase (PI3K), suppressing cellular growth, proliferation, and survival [28,38]. The dominant activation event in the PI3K pathway appears to be *PTEN* protein loss. The PI3K/AKT/mTOR pathway is the most deregulated signalling pathway and is affected in more than 80% of ES-EC [28,38]. McConechy et al., 2012 [38] reported 76.5% in both *PTEN* and *PIK3CA* mutations in ES-EC and their combination is reported to be up to 28.6% in ES-EC. Similarly, Katsutoshi et al., 2005 presented *PTEN* loss of function and *PIK3CA* mutation prevalence, confirming their involvement in activating the PI3K pathway [50]. Cheung et al., 2011 [51] demonstrated that PIK3R1 (p85α) mutations occur at a higher rate in EC than in any other tumour lineage. *PIK3CA* and PIK3R1 mutations are independently linked to favourable survival, although *PIK3CA* is also linked to recurrency in ES-EC [39,52]. On the other hand, PIK3R1 was deemed an unfavourable prognostic factor for ES-EC [25]. Furthermore, Hayes et al., 2006 [39] identified *PIK3CA* as a marker for disease invasion, confirming Samuel et al.’s findings.

#### 2.2.2. *POLE*

*POLE* mutations occur in 7–12% of ECs [29]. *POLE* is responsible for the regulation of glycolysis and cytokine secretion and therefore affects cell metabolism and immune response. Imboden et al., 2019 [52] identified that patients with *POLE*-mutated tumours were significantly younger. In this study, patients with *POLE* mutation appeared to also be nulliparous and to have a history of smoking. The tumours themselves are portrayed to be aneuploidy more frequently. As for prognosis, these patients appeared to have significantly better results and particularly excellent prognoses in cases with hotspot mutations. *POLE* is deemed a good prognostic factor regarding overall survival and has favourable outcomes and therefore is a good indicator for FST [35,36,37,38]. Haruma et al., 2018 [30] demonstrated that *POLE* mutations in EC are associated with a reduced risk of progression-free survival and distant metastases. This was also demonstrated in combination with MSI features (implicating MMRd) [31,32].

#### 2.2.3. *EGFR* and *HER2*

Epidermal growth factor receptor (*EGFR*) is a family of receptors (HER1, *HER2*, and HER4) that are frequently implicated in EC due to their strong association with the PI3K/AKT and RAS/RAF/MEK pathways [53]. *EGFR* was found in 43–67% of patients with EC and was also linked with shortened disease-free and overall survival [53]. More specifically, *HER2* gene amplification and receptor overexpression was demonstrated in EC. High *HER2* expression is an independent prognostic factor influencing progression-free and overall survival in ES-EC [40,41]. Over-expression of *HER2* was more common among more aggressive cancers with a significantly worse prognosis [41]. After a median follow-up of 50 months, there were 43 (25.4%) recurrences, of which the majority of recurrences were in the *HER2*-positive cohort (50.0%). *HER2* is also linked to an increased chance of recurrence. However, Morrison et al., 2006 [40] reported *HER2* to play a minor role in ES-EC, which is more common in the clinical setting. 

Among the *EGFR* family of receptors, HER-3 expression was not identified in EC [54]. The *EGFR* family was identified in 39.7% of patients, with HER4 being the majority of expressions (49.2%) and *HER2* at 41.3%. However, HER-2 was reported as a more significant prognostic factor in ES-EC than HER4, and it was also associated with high MLH1 expression [55].

#### 2.2.4. *CTNNB1*

*CTNNB1* mutation is found in 20–40% of cases of ES-EC. Imboden et al., 2020 [26] report this mutation to be up to 50% in FIGO I, grades 1 and 2, which is almost double that of *PTEN* mutations (27%). It is a vital component of E-cadherin, which is responsible for cell adhesion and is closely associated with the Wnt signalling pathway. Activation of the Wnt pathway contributes to the progression of tumours, abnormal proliferation, and gene expression. *CTNNB1* mutations were also reported in dual mutation co-operativity such as with *PTEN* loss and also accompanied by a KRAS mutation [34]. However, *CTNNB1* mutations were less commonly present in MSI positive tumours than other genetic alterations [56]

Myers et al. revealed that patients with *CTNNB1* mutation have a risk of recurrence which is nine times higher than in those without mutation [57]. Moreover, Kurnit et al., 2017 [34] found the presence of *CTNNB1* mutations at ES-EC to be associated with higher rates of disease recurrence, lower rates of deep myometrial invasion, and lymphatic/vascular space invasion. Additionally, Stelloo et al., 2016 [35] characterise *CTNNB1* mutations as a more aggressive subset in ES-EC, with 35% of intermediate features in high-risk patients. 

#### 2.2.5. KRAS

KRAS is an inactivated oncogene involved in signal transduction by communicating with several cell membrane receptors, including *EGFR* [58]. Current evidence correlates KRAS mutations with the down-regulation of the MAPK and PI3K/AKT pathways as well as the up-regulation of endometrial cell oestrogen receptors, leading to excessive cell proliferation and carcinogenesis [15,59] and increased cell proliferation and apoptosis.

KRAS has consistently appeared in several studies and is reported to have a relatively high prevalence in ES-EC. KRAS mutations were detected in 10–30% of EC and 6–16% of cases of endometrial hyperplasia [60,61]. Byron et al., 2012 reported KRAS mutations to be as high as 19% in ES-EC [54]. Furthermore, KRAS mutations were found to be linked with MSI-positive EC. On the other hand, no association was found with *FGFR2* and *CTNNB1* mutations [45,56].

KRAS mutation can cause hypermethylation changes in genome expression. More specifically, Muraki et al., 2019 found hypermethylation of the MLH1 promoter in 40% of ES-EC cases, which can cause concurrent loss of function in DNA repair proteins [36,62].

Wang et al., 2012 [42] reported KRAS to have significant effects on the recurrence of EC individually as well as when combined with *PIK3CA* [25]. KRAS mutations were also associated with longer disease-free survival.

#### 2.2.6. *FGFR2*

*FGFR2* mutations were reported independently in EC [45]. Studies demonstrate a similar link between somatic oncogenic *FGFR2* mutations in EC as with cervical cancer. Dutt et al., 2008 [46] reported *FGFR2* somatic mutations in 12% of EC samples. Furthermore, it was found that patients with *FGFR2* mutation had shorter progression-free survival and EC-specific survival [63]. Gatius et al., 2011 [45] showed that ES-EC has a higher expression of *FGFR2* than non-endometrioid EC. In the early stages of the disease, *FGFR2* mutations were correlated with shorter disease-free and overall survival [62]. However, Pollock et al., 2007 [47] identified no association between *FGFR2* mutation and overall or disease-free survival in ES-EC.

*FGFR2* immunostaining was statistically significantly associated with estrogen and progesterone receptors and inversely associated with *PTEN* expression. Additionally, *FGFR2* mutations coexisted with mutations in *PTEN*, *PIK3CA*, and CTNNB-1. *FGFR2* mutations were also significantly more common in MSI-positive tumours than *CTNNB1* mutations and appeared to have shorter disease-free survival [45].

#### 2.2.7. *ARID1A*

*ARID1A* was reported in 19–44% of ECs [42]. *ARID1A* (AT-rich interactive domain 1A) is located on chromosome 1p36.11 and encodes ARD1A protein which is a vital component of the SWI/SNF (switch/sucrose non-fermenting) complex. This complex is responsible for regulating proliferation, DNA repair, differentiation, and tumour suppression [64]. Two studies identified ARD1A as a tumour suppressor gene linked to gynaecological diseases and, more specifically, EC [42,65]. Since then, studies reported an increased number of mutations in the *ARID1A* gene in 26–37% of EC-EC. Werner et al., 2013 [42] reported that loss of *ARID1A* is associated with an early event in the process of the carcinogenesis of endometrioid carcinomas and with deep myometrial infiltration. The ARID1A T-rich interactive domain family was also associated with MSI as frequently as 23.1% [9]

#### 2.2.8. *P53*

Suppressor *P53* protein (*TP53*), encoded by the *P53* gene, is highly involved in the cell cycle, differentiation, and apoptosis. *P53* mutations lead to rapid tumour progression and invasion, which is associated with poor ES-EC prognosis [63]. Mutation of the *P53* gene was found in 10–20% of EC, while *TP53* overexpression was present in 20–30%, being a very common abnormality in several human cancers [27,66].

*P53* was described to have poor results in mortality and was linked to both recurrence and metastasis independently as well as when combined with *L1CAM* [27,43,44,66,67]. *L1CAM* is an X-linked genetic mutation located on the Xq28 gene. It encodes for the L1 protein, which spans the cell membrane of nerve cells and allows for neighbouring cell adhesion. L1 protein also plays a role in migration, the organisation of neurons, and axon outgrowth [68]. Additionally, Kommos et al., 2018 [43] identified *L1CAM* expression to be present in 80% of *P53* abnormal tumours. The PORTEC trial found that positive *L1CAM* expression in stage I EC patients had a significant correlation with distant recurrence and overall survival [69]. *L1CAM* expression is, therefore, significantly, but not universally, associated with mutant *P53*. It may be strong enough for clinical implementation as a prognostic marker in combination with *P53* as well as a promising therapeutic target [70].

## 3. Discussion

This is the first thorough systematic review that identifies and summarises the most well-established biomolecular and genetic prognostic factors that facilitate FST decision-making in cases of ES-EC. The classification of biomolecular and genetic prognostic factors as ‘good’, ‘fair,’ and ‘poor’ is an important aspect in the management of patients who wish to risk ES-EC progression for a chance in motherhood. Markers that designate bad prognosis, metastasis, and early recurrency could be used to deny FST, and on the contrary, markers that demonstrate a good prognosis can help clinicians in the management of patients wishing to preserve their fertility. Markers that are fair prognostic factors require further discussion with the patient, discussing the risks. Our review summarises and describes these results and their significance but also indicates the current gaps of knowledge in this field of research.

### 3.1. Fertility Sparing Treatment (FST)

The recurrence rate was reported to be as high as 35%, and patients were advised on the importance of pursuing pregnancy soon after remission and hysterectomy soon after family planning completion [71,72,73]. Kim et al., 2009 [74] reported the recurrence rate after FST to be 38.9% in ES-EC, which is much higher than 5.5% and 5.5% in combined histology EC and G2 EC, respectively. Alternatively, endometrial intraepithelial neoplasia was shown to have a much higher recurrence rate (50%) after FST in a 3 months interval follow-up of endometrial sampling by hysteroscopy. Moreover, Kim et al., 2009 [74] showed that after a median follow-up of 40.7 months, 12 patients (66.7%) preserved their uterus and 8 patients (53.3%) became pregnant with a total of 14 successful pregnancies among patients trying to become pregnant in both groups. Thirty-three percent of patients were reported to have stable disease, and 66.7% had a complete response rate, of which 25% relapsed [75]. Other studies report a relatively high number of foetal losses at 31.3% but also a live birth rate of 72% after FST for EC [76]. However, the limited number of studies describing obstetric outcomes can influence these numbers. It is important to note that there are also clinicopathological factors that can affect FST, such as polycystic ovarian syndrome (PCOS), obesity, diabetes, anovulation, exogenous oestrogen exposure, nulliparity, amenorrhea, and irregular menstruation [11,77,78]. More specifically, in patients with PCOS and reproductive failure, metformin administration and vitamin D supplementation with inositol successfully improved ovulation restoration [79,80,81]. This was especially true in pregnancy where, due to insulin resistance, patients tend to develop gestational diabetes [82].

### 3.2. Eligibility Criteria for FST

When considering a conservative management approach in ES-EC, we should consider the clinical and pathological characteristics of the tumour in order to select the appropriate medical intervention. A conservative management approach could be considered in patients < 40 years old who intend to preserve fertility and plan to conceive as soon as possible after remission. They should have no contraindications for medical treatment and a histological diagnosis of grade I EC; histotype: endometrioid with positive hormone receptor (type I), tumour diameter < 2.0 cm, stage IA without myometrial and adnexal involvement, negative lymph-vascular space invasion, and diffuse immunohistochemical expression of progesterone receptors on endometrial biopsy. These are the patients who are considered to be at “low risk” [83]. Furthermore, according to the Gynecologic Oncology Group (GOG) and Federation International of Gynecologic and Obstetrics (FIGO), the most important prognostic factors for lymph node metastasis in patients with EC were the grade of tumour and the depth of myometrial invasion with the risk of involvement less than 1% and excellent 5 year progression-free survival of 95% if the tumour is grade 1 with overall survival of 90%. In the absence of these risk factors, a conservative approach to surgical staging is feasible, safe, and not associated with an increase in cancer-related mortality [84,85].

### 3.3. Biomolecular and Genetic Prognostic Factors Discussed

#### 3.3.1. *PTEN*

The use of *PTEN* alteration as a prognostic factor is still controversial. Studies found *PTEN* mutations to be associated with favourable clinical and pathologic characteristics, while *PTEN* promoter methylation and *PTEN* loss of function were linked with poor prognosis and metastatic disease [19,24,86]. On the one hand, it is suggested that *PTEN* may be a tumour cell regulator for invasion and metastasis, but on the other hand, it is suggested that *PTEN* inactivation by mutation is an early event in endometrial tumourigenesis and therefore not linked to the metastatic progression of the disease [24,48,86]. *PTEN* alterations were linked to advanced disease in cancers other than EC but rarely presented in gynaecological cancers other than EC [24,87].

Studies showed that *PTEN* mutations occur in the earliest stages of EC and frequently coexist with other mutations [22,46]. *PTEN* loss of function is one of the most frequently identified mutations in ES-EC and negatively affects the regulation of the PI3K-AKT. In fact, *PTEN* and PI3K/Akt/mTOR signalling pathways were associated with poor prognosis [88,89,90]. Interestingly, these mutations were reported in more cases of EECs (75%) than in non-EECs (43%) [91].

To conclude, *PTEN* loss is overall a good prognostic factor for ES-EC. However, these findings are inconsistent among the literature, and therefore large clinical trials are required to examine its effectiveness as a prognostic factor for FST in patients with ES-EC. Additionally, the accurate and fast identification of this mutation is vital given the narrow time window that the clinicians have to decide patient eligibility for FST in order to achieve optimal therapeutic benefit. Djordjevic et al., 2012 [91] demonstrated that *PTEN* immunohistochemistry is a more effective tool in detecting the majority of cases with *PTEN* loss of function in a quick and cost-effective manner compared to *PTEN* sequencing.

#### 3.3.2. MSI and MMR

MMR genes work with the DNA repair system to promote genetic stability. The MMR system plays a core role in carcinogenic mechanisms of ES-EC. These actions cause oncogene mutation, inactivation of tumour suppressor genes, and oncogene activation leading to chaotic cell proliferation and, consequently, carcinogenesis.

Mutations during DNA replication and defects in the MMR genes result in MSI. MSIs have a predictive value for the efficacy of immune checkpoint inhibitors in metastatic tumours regardless of primary tissue origin. This was initially discovered in HNPCC patients along with MMR mutations [92].

MSI is a useful biomarker for identifying patients who have a good prognosis [30]. However, results on MSI as a prognostic factor are inconsistent. It is linked to recurrence but not to metastasis and overall survival. We can therefore argue that MSI is a fair prognostic factor for FST [30]. This inconsistency is based on a number of studies showing insignificant results rather than contradictory data on the effect of MSI in ES-EC. Testing for MMR status/MSI in ES-EC is of vital importance, and it is also identified in patients who are at higher risk for human non-polyposis colorectal cancer (HNPCC/Lynch Syndrome) [17]. Testing for EC in patients with HNPCC is therefore advised, even though currently there is limited evidence on the benefits of HNPCC-associated EC screening.

HNPCC-associated EC cases lack additional mutations, suggesting that in the MMRd context, few additional molecular changes lead from pre-invasive lesions to carcinoma [93]. For these patients, hysterectomy and bilateral salpingo-oophorectomy might be a more appropriate treatment method than FST as a preventative measure for both endometrial and ovarian cancer. This should preferably be before the age of 40 years [94].

Travaglino et al. also showed Dusp6, a MAPK signalling pathway marker, to be an indicator of good response to treatment along with the deficiency of MMR [88]. This was also true when combined with *PTEN*. The combination of *PTEN* involvement, MMRd, and Dusp6 deficiency were proven to be important prognostic factors in conservative treatment failure.

However, evidence from MMRd studies is inconsistent. MMR status is suggested to be a predictive biomarker for FST. It is described as not associated with disease progression, but at the same time, it is linked to have a relatively high rate of recurrence. Chung et al., 2021 identified 9 patients with MMRd among a cohort of 54 (17%). Four of these patients (44%) underwent immediate hysterectomy because of FST failure, and three patients (33%) presented with an upstaged diagnosis after hysterectomy [33]. However, this study had a relatively small cohort, and for validity, larger-scale studies are required.

In conclusion, the International Society of Gynecological Pathology recommended using MMR-immunohistochemistry (IHC) testing for both MMR status and MSI in all EC samples, irrespective of patient age [95]. Using IHC, the expression of four MMR proteins MLH1, PMS2, MSH6, and MSH2, are assessed, and additionally, PMS2 and MSH6 antibodies can also be assessed [96].

#### 3.3.3. *POLE*

The *POLE* gene encodes the major catalytic subunit of DNA polymerase-ε [51]. Polymerase-ε is thought to function in strand synthesis [13]. *POLE* mutations improve the prognosis of EC by regulating cellular metabolism through AMF/AMFR signal transduction [31]. Li et al. identified both AMF/PGI and AMFR/gp78 to have higher expression in *POLE* mutants [31]. Comprehensive low expression of *POLE* and high expression of AMFR/gp78 showed a positive correlation with patient survival time. Phosphoglucose isomerase (PGI) is a glycolytic enzyme involved in the gluconeogenesis–glycolysis pathways. It is an extracellular cytokine as well as an autocrine motility factor (AMF). Therefore, AMF/PGI plays a dual role as a phosphor-glucose isomerase that catalyses the interconversion of glucose-6-phosphate and fructose-6-phosphatein glycol metabolism when it is effective as a cytokine [31,53,97]. Furthermore, *POLE* mutations are also linked to ultrahigh mutation rates and frequent activation of WNT/*CTNNB1* signalling. Li et al., 2019 [31] showed that the presence of *POLE* mutations in the early clinical stage (I + II) and low histologic grade (G1) EC had a favourable prognosis. The main reason suggested was the somatic *POLE* ultra-mutation which causes an abundance of antigenic neoepitopes that triggers an anti-tumour immune response [98]. Stello et al., 2016 [35] also described favourable features in 50% of *POLE*-mutant EC in the absence of MSI and *CTNNB1* mutations.

Haruma et al., 2018 [30] demonstrated that ECs *POLE* mutations are associated with a reduced risk of recurrence and distant metastases. This was also demonstrated in combination with MSI features (implicating dMMR). Despite *POLE* mutations being considered a good prognostic factor, favouring FST, Veneris et al. [29] analysed a case where recurrence was observed, concluding that *POLE*-mutated EC has high tumour mutation burden, tumour neoantigen production, and tumour-infiltrating T cells. In addition, Van Gool et al. reported *POLE* mutations in 7–12% of EC and demonstrated that *POLE*-mutant ECs have an increased lymphocytic infiltrate in comparison to *POLE* wild-type/MSI-high and *POLE* wild-type/MSS subgroups [70]. However, these studies include a low number of participants with complex EC histology.

Imboden et al., 2019 [52] concluded that the *POLE*-mutated EC definition needs further specification to achieve a more accurate report on survival prognosis. This requires the inclusion of clinicopathologic characteristic variants of uncertain significance such as parity, BMI status, and smoking status. *POLE*-mutated EC was linked, though not statistically significant, to nulliparous women with lower BMI and often current or past smokers.

Therefore, despite the fact that *POLE* mutations can significantly improve ES-EC prognosis, in order to be eligible for FST, additional genetic alterations specific to the patient’s characteristics need to be considered.

#### 3.3.4. *EGFR*, *HER2*

*HER2* is linked to poor overall survival in EC. Morrison et al., 2006 [40] reported a median overall survival of 5.2 years for patients with overexpression of *HER2*, 3.5 years for patients with expression of *HER2*, and 13 years for patients who did not express *HER2* on their cancers. Okuda et al., 2010 [19] reported *HER2* overexpression to be more frequent in non-EECs and suggested that *HER2* overexpression in ES-EC characterises late progression and differentiation events. A small cohort study also confirmed increased expression of *EGFR* and *HER2* overexpression [53]. Lastly, Erickson et al., 2020 [41] reported *HER2*-positive tumours to have worse progression-free survival, recurrence, and overall survival, after a median follow-up of 50 months in 169 stage I uterine serous carcinomas.

*EGFR* receptors have a vital role in the carcinogenesis of EC. More specifically, *HER2* and HER4 overexpression is linked to more aggressive, high-grade ECs and indicate a poor prognosis. In ES-EC, the evidence is limited, but *EGFR* and *HER2* are so far consistently considered poor prognostic factors for ES-EC. FST is therefore not recommended for these patients as more harm than good might be seen from delaying treatment [41].

#### 3.3.5. *CTNNB1*

Patients with *CTNNB1* somatic gene mutation appear at high incidence in ES-EC. The accumulation of beta-catenin was inversely correlated with the patient’s age [89]. In this setting, the *CTNNB1* mutation has a major role as a molecular classifier of EC, especially in young patients [33]. Following FST with progesterone, the expression of β-catenin was significantly increased in patients with disease progression [90].

When it comes to recurrency, *CTNNB1* is described as a fair prognostic factor in ES-EC, even though the results are not always significant. Additionally, Kurnit et al., 2017 [34] found *CTNNB1* mutations to be risk factors for disease recurrence even in presumed low-risk patients. These patients are usually of a younger age, at an early stage of the disease, and have a low incidence of lymphatic vascular space. Although Hu et al., 2019 [90] did not identify it as a marker on recurrence, they describe that due to its high prevalence in ES-EC, it can play a role in pathogenesis and early treatment.

Further research on *CTNNB1* mutation-related ES-EC and its prognosis after FST are essential in identifying its clinical relevance in decision making.

#### 3.3.6. KRAS

Although evidence is limited and occasionally conflicting, there is a clear trend in the literature showing that KRAS plays a role early in EC progression, especially when the disease originates from hyperplastic endometrium [99].

Cote et al., 2012 [37] exclusively studied African American patients and significantly associated KRAS as a mutation that commonly occurs in ES-EC. Additionally, there were no observed mutations in other histological EC tumours such as serous, clear cell, or mucinous tumour types arguing that KRAS mutations may not have metastatic potential. However, so far, metastasis in KRAS mutations has not been studied [37].

KRAS is so far defined as a good prognostic indicator in mortality but has proven to have a high prevalence in recurrency. Data on metastasis is not yet available in literature, and KRAS can therefore be used as a fair prognostic factor in ES-EC favouring FST [59]. However, the results are relatively inconsistent, and the cohorts are too small to be able to validate their significance in the literature [25,37].

#### 3.3.7. *FGFR2*

Although KRAS and *FGFR2* mutations share similar activation of the MAPK pathway, Byron et al., 2012 [62] and Jeske et al., 2014 [63] suggest very different roles in tumour biology. Furthermore, Jeske deemed a significantly higher relative risk of failure and shorter progression-free survival when known clinicopathological factors such as age, stage, and grade when taken into consideration.

Jeske and colleagues showed *FGFR2* mutation as more prevalent among advanced age (≥70 years) patients. *FGFR2* mutations were also consistent with more aggressive disease *and* were more common in patients with more advanced stage III/IV, although this did not reach statistical significance [63].

The identification of activating mutations in *FGFR2* in ES-EC is of direct clinical relevance, and further studies are required to identify the relationship with recurrence, metastasis, and overall survival.

#### 3.3.8. *ARID1A*

*ARID1A* RNA expression is significantly correlated with *ARID1A* protein loss. Thus, loss of *ARID1A* appears to be an early event in the carcinogenesis of endometrioid uterine carcinomas, and the association with deep myometrial infiltration may suggest importance for invasiveness. Werner et al., 2013 [42] identified *ARID1A* loss to be associated with younger patients and diploid tumour cells, suggesting *ARID1A* loss relationship with less aggressive EC. Lastly, the evidence in the literature is inconsistent, and there is no relationship status between *ARID1A* loss and disease progression or overall survival.

#### 3.3.9. *P53*

*TP53* mutation is described as one of the most important molecular factors which predict prognosis in ES-EC and is associated with an unfavourable outcome [19,56]. Levine et al. reported fewer *TP53* mutations in ES-EC compared to more frequent mutations (*PTEN*, *CTNNB1*, *PIK3CA*, *ARID1A*, *KRAS*). Sherman et al., 1995 [16] debated that in endometrial intraepithelial carcinoma, as well as in transformation and dedifferentiation of other neoplasms, *P53* protein expression plays a significant role. Therefore, *L1CAM* status presence among *TP53* mutations can be a significant prognostic factor for worse disease-specific survival. This was described consistently over the literature, and we can therefore define *TP53*, as well as its combination with *L1CAM*, as poor prognostic factors for ES-EC and subsequently for FST.

### 3.4. Fertility Sparring Treatment in Current Clinical Practice

The European Society of Gynecological Oncology (ESGO) confirm that FST is a safe option for patients with ES-EC (Stage1A with endometrial histological type and grade 1 EC) [100].

A Swedish nationwide population-based cohort study identified that natural fertility was maintained after FSS in all patients with 11% of women giving birth to healthy children, all delivered at full-term [101]. Additionally, complete and partial response to progestin-based FST was identified to be up to 83% in patients with ES-EC. Relapse was diagnosed in 20% of these patients, with the total number of pregnancies at 43% and total live births at 30% [102]. In a systematic review and meta-analysis by Gallos et al., 2012 28% of women with ES-EC progressed to have live births following FST. Similarly, Cappelletti et al., 2021 suggested that progestin-based FST is viable for women with well-differentiated, clinical-stage 1A, EEC. Even though only one out of five women were estimated to achieve a live birth, the use of prognostic factors can improve both patient treatment selection and reproductive outcomes [103].

### 3.5. Limitations

To date, there are limited articles available on the oncological safety of FST as well as a limited number of studies specific to ES-EC. The papers that are specific do not always look into individual genetic mutations but in groups of several genetic mutations. Some of these genetic alterations are already established in clinical practice, but a lot of them are novel and not reliably tested through clinical trials. The literature mainly consists of small cohorts, retrospective case series, and animal studies. Animal tissue studies were not deemed appropriate for this study and were therefore excluded.

The genes tested through large cohort studies were compared to cancers other than ES-EC and therefore affected by genetic factors which may not be solely impactful in ES-EC. Additionally, the cohort of patients tested in ES-EC is smaller compared to the cohorts used to test genetic mutations in other cancers such as ovarian and colorectal cancer. Follow-up of these patients is often short, and incidence of pregnancy and pregnancy outcomes are inadequately reported. Consequently, FST uncertainty in patients with ES-EC is high and prognostic factors favouring FST for women with Stage IA Grade 1,2 EC (ES-EC) are limited.

## 4. Materials and Methods

### 4.1. Inclusion Criteria

The articles were screened to check that they met the following criteria: early stage, low-grade, endometrial cancer patients, reporting recurrence, metastasis, overall survival, obstetric outcomes, or progression-free survival, reporting prognostic genetic or biomolecular markers, and in the English language. All studies on animals or studies involving ex situ tissues were excluded.

### 4.2. Search Strategy

A comprehensive search strategy was carried out using the NICE Healthcare Databases Advanced Search (National Institute of Health and Clinical Excellence). Four databases, Cochrane, Embase, MEDLINE, and PubMed, were searched between March 1946 and 22nd December 2022. The following search strategy was used: (‘’Fertility sparring treatment.mp. OR Fertility sparing surgery.mp. OR Conservative management.mp. OR Conservative Treatment’’) AND (‘’Early stage endometrial cancer.mp. OR low grade endometrial cancer.mp. OR Stage IA Grade 1 endometrial cancer.mp. OR Stage IA Grade 2 endometrial cancer.mp.’’) AND (‘’biomolecular prognostic factors.mp. OR genetic prognostic factors.mp. OR biomolecular markers.mp. OR Genetic Markers OR estrogen receptor*.mp. OR progesterone receptor*.mp. OR estrogen marker*.mp. OR progesterone marker*.mp.’’)

This yielded 29 results across the 4 databases. This was reduced to 26 after duplicates were removed. The titles and abstracts were screened by the first 2 authors (P.T. and S.D.), and 20 potentially relevant articles were found. Of the remaining studies, the full manuscripts were reviewed by the first 2 authors to ascertain whether the inclusion criteria were met. If there was disagreement between the first 2 authors regarding a study, the matter was referred to the most senior author (V.T.). Consequently, 18 articles were selected for inclusion in the review. The bibliography of these manuscripts was then independently screened by the first 2 authors, searching for any other potentially relevant studies. Twenty further studies were found by this method, bringing the final total to thirty-eight studies, of which thirty-four were finally unanimously agreed to be included (Figure 3). This systematic review was registered with PROSPERO (CRD42022312003) and is in line with the PRISMA criteria checklist [51].

### 4.3. Data Extraction

Once the studies were selected, the manuscripts were reviewed independently by the first 2 authors. The primary objective was to collect genetic and biomolecular prognostic factors of ES-EC. The secondary outcome was to identify which of these molecular mechanisms had an impact on functional and clinical outcomes at the end of a follow-up period. Outcomes were measured in accordance with recurrence, metastasis, overall survival, progression-free survival, or obstetric outcomes. Patient demographics, the number of tissues tested, genetic identification technique, adverse events, treatment failure, and details of concomitant therapies were also recorded when available. The final 34 articles encompassed 9165 patients (Table 2).

### 4.4. Methodological Quality Assessment

Two authors (P.T. and S.D.) independently assessed the methodological quality of each study using the Critical Appraisal Skills Program (CASP) to increase the rigour of this review. This allowed for a structured approach in assessing the results and their clinical relevance. The following domains were assessed to see whether the criteria were “Clearly met” (+) or “Clearly not met” (−). “Cannot tell” (?) was used to describe cases in which the authors were not able to assess whether criteria were met. If there was disagreement between the authors, then the senior author (V.T.) was consulted, and disagreement was resolved by consensus.

## 5. Conclusions

*PTEN*, *PIK3CA*, KRAS, *FGFR2*, *CTNNB1*, MSI, and ARIDIA mutations are linked to good five-year survival (85%) for EC, and *TP53* is labelled as a poor prognostic factor with 55% five-year survival for EC [78]. However, this data is not ES-EC specific. At the reproductive stage, where ES-EC is the most common clinical presentation of EC, the data is still inconsistent and not universally agreed upon. After recurrence rate, risk of metastasis, and mortality were considered; *PTEN* and *POLE* alterations were found to be good prognostic factors of ES-EC, favouring FST. MSI, *CTNNB1*, and *K-RAS* alterations were found to be fair prognostic factors of ES-EC, favouring FST, but have a higher risk of recurrence. *PIK3CA*, *HER2*, *ARID1A*, *P53*, *L1CAM*, and *FGFR2* were found to be poor prognostic factors of ES-EC and, therefore, not favouring FST (Figure 2). However, in the decision-making process, patients’ clinicopathological characteristics have to be taken into consideration. Interestingly, currently there are numerous ongoing clinical trials that investigate different types of FST (NCT01594879, NCT03241914, NCT02990728, NCT03463252, NCT03538704, NCT04362046) but there are no current clinical trials focusing on patient treatment selection using genetic prognostic factors. In the future, larger clinical trials and studies with bigger cohorts will be needed to confidently choose FST for the treatment of ES-EC from favourable prognostic factors.

## Figures and Tables

**Figure 1 ijms-23-02653-f001:**
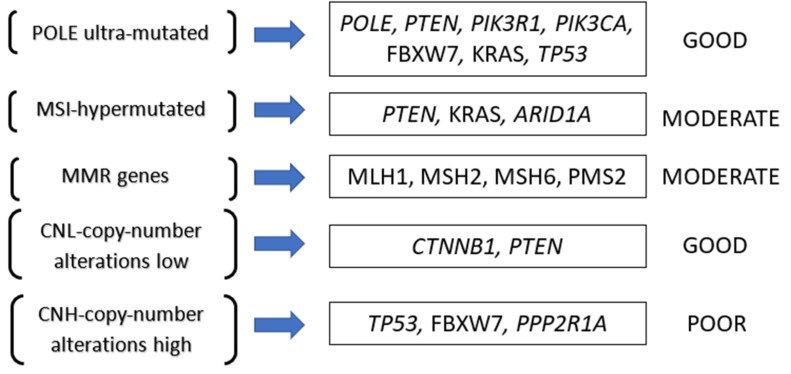
Prognostic categories as established by the Cancer Genome Atlas Research Network (CGARN).

**Figure 2 ijms-23-02653-f002:**
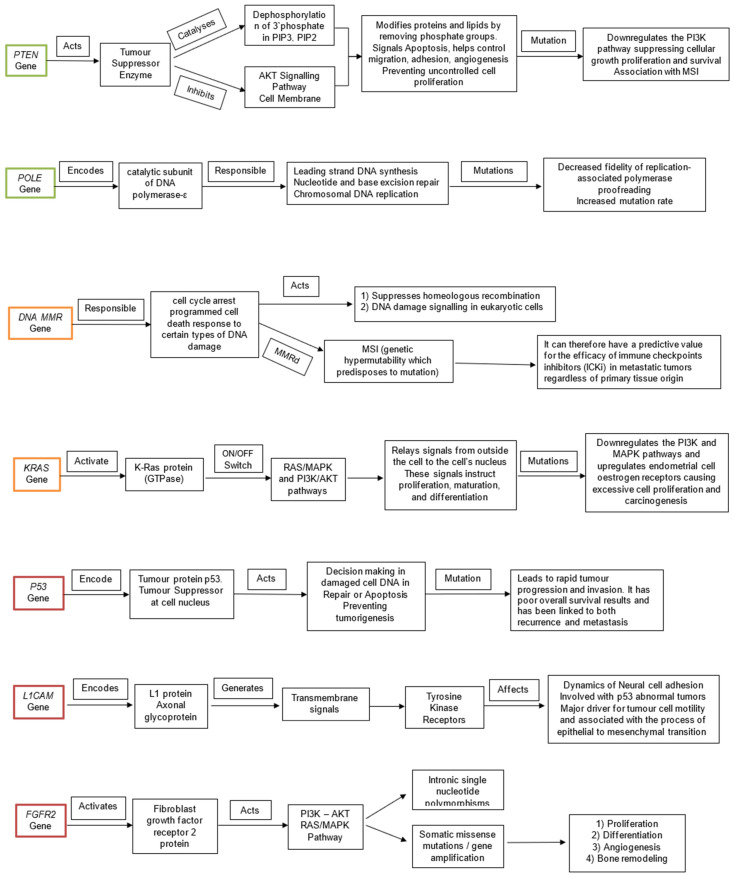
Visual representation of the functions of each gene (*PTEN, POLE, MMR, KRAS, P53, L1CAM, FGFR2*) and the pathological consequences of their mutations. The genes are colour coded according to prognostic ability good (green), fair (amber), poor (red).

**Figure 3 ijms-23-02653-f003:**
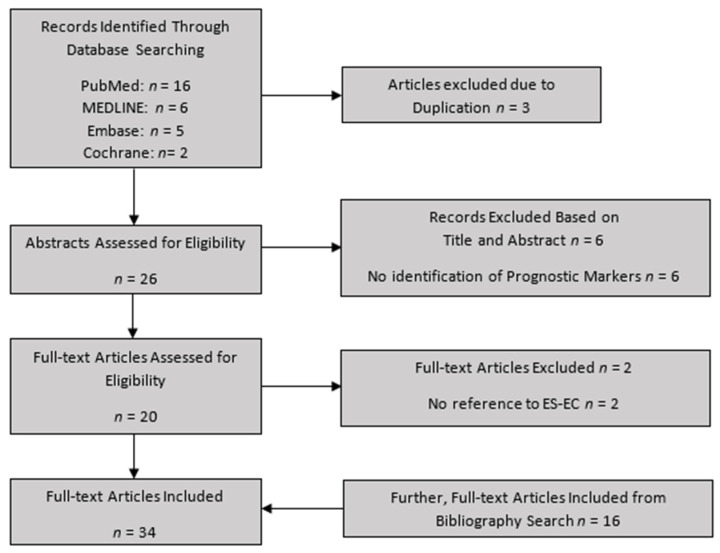
Flowchart for study selection.

**Table 1 ijms-23-02653-t001:** Detailed analysis of prognostic factors in early stage endometrial cancer (ES-EC).

References	Biomarkers	Mutation	Mortality (3 Points)	Metastasis (2 Points)	Recurrency (1 Point)	Total Points	Prognosis	Level of Evidence
Salvesen et al., 2004 [24] Wang et al., 2012 [25] Imboden et al., 2020 [26] Cavaliere et al., 2021 [27]	*PTEN*	Point mutations or deletions	No	No	N/A	0	Good	Inconsistent
Levine et al., 1998 [28] Veneris et al., 2019 [29] Haruma et al., 2018 [30] Li et al., 2019 [31] Imboden et al., 2020 [26]	*POLE*	Somatic missense mutations	No	No	No	0	Good	Inconsistent
Parc et al., 2000 [32] Haruma et al., 2018 [30] Chung et al., 2021 [33]	MSI	Mismatched repairs insertion/deletion mutations	No	No	Yes	1	Fair	Inconsistent
Imboden et al., 2020 [26] Kurnit et al., 2017 [34] Stelloo et al., 2016 [35] Cavaliere et al., 2021 [27]	*CTNNB1*	Duplication or deletion	No	No	Yes	1	Fair	Consistent
Wang et al., 2012 [25] Sideris et al., 2019 [36] Cote et al., 2012 [37]	*K-RAS*	Activating mutations	No	N/A	Yes	1	Fair	Consistent
McConechy et al., 2012 [38] Hayes et al., 2006 [39]	*PIK3CA*	Activating mutations	No	Yes	Yes	3	Poor	Consistent
Morrison et al., 2006 [40] Erickson et al., 2020 [41]	*EGFR* *HER2*	Activating mutations overexpression	Yes	N/A	Yes	4	Poor	Inconsistent
Werner et al., 2013 [42] Kommos et al., 2018 [43]	*ARID1A*	Frameshift or nonsense mutations	N/A	Yes	Yes	3	Poor	Inconsistent
Cavaliere et al., 2021 [27] Chung et al., 2021 [33]	*P53*	Point mutations	Yes	Yes	Yes	6	Poor	Consistent
Cavaliere et al., 2021 [27] Kommos et al., 2018 [43] Smogeli et al., 2016 [44]	*L1CAM*	X-linked mutation	Yes	Yes	Yes	6	Poor	Consistent
Gatius et al., 2011 [45] Dutt et al., 2008 [46] Pollock et al., 2007 [47]	*FGFR2*	Point mutations	Yes	N/A	N/A	3	Poor	Consistent

No = favourable survival, not linked to recurrency of disease or metastasis, Yes = poor overall survival, linked to recurrence or metastatic disease, N/A = information was not available in the literature. Scoring Prognosis by using point system: Good = Total Points 0, Fair = Total Points 1, Poor = Total Points 2–6. Level of evidence is deemed Consistent and Inconsistent according to literature reports. *PTEN*; phosphatase and tensin homolog, *POLE*; polymerase epsilon, *HER2*; Human epidermal growth factor receptor 2 MSI; microsatellite instability, KRAS; Kirsten rat sarcoma viral oncogene homolog, MMR; mismatch repair; *P53*; tumour protein 53.

**Table 2 ijms-23-02653-t002:** Quantitative data of the final 34 Studies included in the review.

34 Quantitative Studies	Total Patients (*n* = Number)
Endometroid Endometrial Cancer (EEC)	*n* = 9165
Early Stage Endometrial Cancer (ES-EC)	*n* = 4097
Grade 1	*n* = 572
Grade 2	*n* = 286
FIGO Stage 1	*n* = 649
FIGO Stage 2	*n* = 137

## Data Availability

Not applicable.

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
