# Peer review of "Biomolecular and Genetic Prognostic Factors That Can Facilitate Fertility-Sparing Treatment (FST) Decision Making in Early Stage Endometrial Cancer (ES-EC): A Systematic Review"

_ijms, 2022, doi:10.3390/ijms23052653_

Round 1
Reviewer 1 Report
In the present paper the authors investigate the biomolecular and genetic prognostic factors impact of early-stage endometrial cancer patients. It is a very interesting and exciting retrospective analysis. The manuscript is well written and structured. Documentation of the results well supports points of discussion as well as conclusions. This article investigated theoretical effectivity and significance of the clinical impact of biomolecular and genetic prognostic factor in early-stage endometrial cancer patients.
There is only one point that need to be strengthened the discussion may not so related to fertility sparing treatment, some modification may be needed to emphasis this part.
Author Response
Responses to comments made by Reviewer #1
1) In the present paper the authors investigate the biomolecular and genetic prognostic factors impact of early-stage endometrial cancer patients. It is a very interesting and exciting retrospective analysis. The manuscript is well written and structured. Documentation of the results well supports points of discussion as well as conclusions. This article investigated theoretical effectivity and significance of the clinical impact of biomolecular and genetic prognostic factor in early-stage endometrial cancer patients.
Response: We would like to thank Reviewer #1 for reviewing our paper and recognizing the benefits of our work upon the existing literature.
2) There is only one point that need to be strengthened the discussion may not so related to fertility sparing treatment, some modification may be needed to emphasis this part.
Response: We would like to thank Reviewer #1 for their comments. We agree and therefore have modified our discussion and added a new paragraph which includes the current application of FST in clinical practice with the aim of strengthening the message we are trying to convey to the reader.
‘’ The European Society of Gynecological Oncology (ESGO) confirm that FST is a safe option for patients with ES-EC (Stage1A with endometrial histological type and grade 1 EC) [97]. A Swedish nationwide population-based cohort study identified that natural fertility was maintained after FSS in all patients with 11% of women giving birth to healthy children, all delivered at full-term [98]. Additionally, complete and partial response to progestin-based FST was identified to be up to 83% in patients with ES-EC. Relapse was diagnosed in 20% of these patients with the total number of pregnancies at 43% and total live births at 30% [99]. In a systematic review and meta-analysis by Gallos et.al. 2012, 28% of women with ES-EC progressed to have live births following FST. Similarly, Cappelletti et.al. 2021 suggested that progestin-based FST is viable for women with well-differentiated, clinical stage 1A, EEC. Even though only one out of five women were estimated to achieve live birth, the use of prognostic factors can improve both patient treatment selection and reproductive outcomes [100].’'
Page 14,15 Section 4.4. Fertility Sparring Treatment in current clinical practice
Reviewer 2 Report
Summary:
The manuscript is a review of 34 studies with 9165 patients regarding Fer-tility Sparing Treatment (FST) in Early-Stage Endometrial Cancer (ES-EC)
Strengths:
- The review presents biomolecular and genetic factors in various categories e.g. good prognostic factors of ES-EC, favouring FST, fair prognostic factors of ES-EC, favouring FST, but with risk of recurrence, poor prognostic factors of ES-EC, not favouring FST
- Various important genes associated with cancer and cell cycle are classified into one of the above categories, e.g. PTEN, POLE, MSI, CTNNB1, K-RAS, PIK3CA, HER2, ARID1A, p53, L1CAM, FGFR2
Weaknesses:
The following concerns need to be addressed.
- Abstract: 1st sentence - "big challenge" - reason needs to be mentioned, frequency if available.
- Abstract concluding sentence undermines in the importance of their own study. The advantage of the study needs to be focused in the last sentence, with a mention of additional need for bigger cohorts.
- Figure 3 legend - POLE and KRAS order is flipped
- Figure 3 - presentation can be improved to make it appear less cluttered. One of the ways is color-coding the 1st box containing name of each gene.
- Clinical trials are mentioned at multiple locations in the text including Conclusion. In the Conclusion section, NCT numbers for a few clinical trials may be included if they are relevant to ES-EC and FST, involving the biomarkers discussed in this manuscript.
Author Response
Responses to comments made by Reviewer #2
1) Strengths: The review presents biomolecular and genetic factors in various categories e.g. good prognostic factors of ES-EC, favouring FST, fair prognostic factors of ES-EC, favouring FST, but with risk of recurrence, poor prognostic factors of ES-EC, not favouring FST
Various important genes associated with cancer and cell cycle are classified into one of the above categories, e.g. PTEN, POLE, MSI, CTNNB1, K-RAS, PIK3CA, HER2, ARID1A, p53, L1CAM, FGFR2
Response: We would like to thank Reviewer #2 for reviewing our paper and recognizing the strengths of our work upon the existing literature.
2) Weaknesses: The following concerns need to be addressed.
Response: We would like to thank Reviewer #2 for reviewing our paper. We would like to thank you for raising these important points which need to be addressed. Please find below how we tackled each point.
2a) Abstract: 1st sentence - "big challenge" - reason needs to be mentioned, frequency if available.
Response: We would like to thank Reviewer #2 for their comments. We have updated the abstract’s introduction and added frequency to make our point clearer and stronger.
‘’Endometrial cancer occurs in up to 29% of women before 40 years of age. Seventy percent of these patients are nulliparous at the time. Decision making on fertility preservation in Early-Stage Endometrial Cancer (ES-EC) is therefore a big challenge since the decision between the risk of cancer progression and a chance to parenthood needs to be made.’’
Page 1 Section Abstract, (Introduction)
2b) Abstract concluding sentence undermines in the importance of their own study. The advantage of the study needs to be focused in the last sentence, with a mention of additional need for bigger cohorts.
Response: We would like to thank Reviewer #2 for recognizing the importance of our work and for their insightful comment. We have modified our last sentence to portray best the advantages of our study.
‘’ Clinical trials with bigger cohorts are needed to further validate the fair genetic prognostic factors. Using the aforementioned good and poor genetic prognostic factors, we can make more confident decisions on FST in ES-EC.’’
Page 1 Section Abstract, (Conclusion)
2c) Figure 3 legend - POLE and KRAS order is flipped
Response: We have changed the order according to prognostic ability. Thank you for observing this detail.
2d) Figure 3 - presentation can be improved to make it appear less cluttered. One of the ways is color-coding the 1st box containing the name of each gene.
Response: We agree with the reviewer’s comment. We were unaware that colour coding is allowed in such figures. We have modified Figure 3 accordingly to make it clearer and less cluttered and more organized.
2e) Clinical trials are mentioned at multiple locations in the text including Conclusion. In the Conclusion section, NCT numbers for a few clinical trials may be included if they are relevant to ES-EC and FST, involving the biomarkers discussed in this manuscript.
Response: We would like to thank Reviewer 2 for their insightful comments. We have added the following NCT numbers for the reader to be easily directed in the current clinical trials ongoing for FST. However, interestingly, there are no clinical trials focusing on patient treatment selection using genetic prognostic factors or the effects of these prognostic factors on reproductive outcomes.
‘’Interestingly, currently there are numerous ongoing clinical trials which investigate different types of FST (NCT01594879, NCT03241914, NCT02990728, NCT03463252, NCT03538704, NCT04362046) but no clinical trials focusing on patient treatment selection using genetic prognostic factors.’’
Page 15, Section Conclusion
This manuscript is a resubmission of an earlier submission. The following is a list of the peer review reports and author responses from that submission.